# A Case for Library-Level k-Means Binning in Histogram Gradient-Boosted Trees

**Asher Labovich**                                                      *asher_labovich@brown.edu*
*Department of Applied Mathematics, Brown University*

**Reviewed on OpenReview:** *https://openreview.net/forum?id=UaTrLLspJa*

## Abstract

Modern Gradient Boosted Decision Trees (GBDTs) accelerate split finding with histogram-based binning, which reduces complexity from $O(N \log N)$ to $O(N)$ by aggregating gradients into fixed-size bins. However, the predominant quantile binning strategy—designed to distribute data points evenly among bins—may overlook critical boundary values that could enhance predictive performance. In this work, we consider a novel approach that replaces quantile binning with a $k$-means discretizer initialized with quantile bins, and justify the swap with a proof showing how, for any $L$-Lipschitz function, k-means maximizes the worst-case explained variance of Y obtained when treating all values in a given bin as equivalent. We test this swap against quantile and uniform binning on 33 OpenML datasets plus synthetics that control for modality, skew, and bin budget. Across 18 regression datasets, k-means shows no statistically significant losses at the 5% level and wins in three cases—most strikingly a 55% MSE drop on one particularly skewed dataset—even though k-means' mean reciprocal rank (MRR) is slightly lower (0.65 vs 0.72). On the 15 classification datasets the two methods are statistically tied (MRR 0.70 vs 0.68) with gaps ≤0.2 pp. Synthetic experiments confirm consistently large MSE gains—typically >20% and rising to 90% as outlier magnitude increases or bin budget drops. We find that k-means keeps error on par with exhaustive (no-binning) splitting when extra cuts add little value, yet still recovers key split points that quantile overlooks. As such, we advocate for a built-in `bin_method=k-means` flag, especially in regression tasks and in tight-budget settings such as the 32–64-bin GPU regime—because it is a "safe default" with large upside, yet adds only a one-off, cacheable overhead ($\approx$ 3.5s per feature to bin 10M rows on one Apple M1 thread).

## 1 Introduction

Gradient Boosted Decision Trees (GBDTs) are ensemble learning methods that have achieved state-of-the-art results on a wide range of tasks. By iteratively fitting new decision trees to the "pseudo-residuals" of a loss function, GBDTs combine many weak learners (e.g. individual decision trees) into a strong predictor (Friedman, 2001). Since the seminal work of Friedman on gradient boosting, the technique has become a *de facto* choice for structured data problems. Because GBDTs require only a twice-differentiable loss function, they are especially useful for non-standard prediction problems, such as ranking or quantile regression (Burges, 2010). Modern implementations like XGBoost, LightGBM, and CatBoost have popularized GBDTs by offering order-of-magnitude computational speed-ups with excellent accuracy (Chen & Guestrin, 2016) (Ke et al., 2017) (Prokhorenkova et al., 2019). These systems have been widely adopted to win Kaggle and other ML competitions, often outperforming deep learning on tabular data (Grinsztajn et al., 2022).

In *formal terms*, GBDT algorithms wish to minimize an arbitrary twice-differentiable loss function by maintaining an ensemble-based predictor $F_t$ comprised of a series of CART trees. At each round t it calculates first and second-order statistics

$$g_i = \frac{\partial L(y_i, F_{t-1}(x_i))}{\partial F_{t-1}(x_i)} \qquad h_i = \frac{\partial^2 L(y_i, F_{t-1}(x_i))}{\partial F_{t-1}(x_i)^2}$$

and then fits a simple CART tree to maximize a split-gain criterion. The optimal weight for leaf $l$ is calculated as

$$w_l = -\frac{\sum_{i \in l} g_i}{\lambda + \sum_{i \in l} h_i}$$

and the model is updated as $F_t(x) = F_{t-1}(x) + \eta w_{leaf_t(x)}$. Parameters $\eta$ and $\lambda$ correspond to the learning rate and L2 regularization respectively, and are chosen by the user prior to training.

A key method for optimizing these algorithms is their use of **histogram-based binning** for splitting continuous features. Traditional decision tree algorithms sort feature values to find split points, which is computationally expensive ($O(N \log N)$ per feature initially, and $O(N)$ for split-finding) and thus intractable for large datasets. To allow training on datasets sometimes too large to fit into memory, modern GBDT algorithms bucket continuous feature values into discrete bins and accumulate gradients per bin during training. This approximation drastically reduces computation and memory overhead while maintaining similar accuracy as exhaustive splitting. In particular, histogram-based models require only $O(N)$ scans for histogram building (and only $O(\# \text{ bins})$ for split-finding), with no sorting needed. Major state-of-the-art algorithms, like XGBoost, LightGBM, and CatBoost, all use similar histogram-binning methods, though with small differences in implementation. These innovations allow GBDTs to effectively scale to datasets with millions or billions of instances.

In state-of-the-art histogram-based GBDT libraries, the default choice is **quantile binning**: thresholds are placed so that every bin holds roughly the same number of observations, barring duplicate values. This equal-frequency distribution is popular because it preserves the rank order of values and often yields split quality comparable to those obtained by exhaustive-search algorithms. However, quantile binning can sometimes mask rare but influential values. For example, in a sample of one million observations with one thousand extreme outliers, a 255-bin quantile scheme would pool those outliers with $\sim 3000$ ordinary points, blunting a potentially important boundary.

A natural alternative is to utilize some method of unsupervised clustering to isolate important points prior to training. One such example is via 1-D $k$-means clustering (Lloyd, 1982), which explicitly minimizes within-bin variance and can place cuts at rare yet influential values. We therefore compare **Lloyd–$k$-means clustering with quantile initialization**[1] with quantile binning as well as uniform (equal-width) binning *to characterize the regimes—tail mass, multi-modality, and bin budget—*under which each discretizer is preferable and to test whether a simple $k$-means swap can boost accuracy without compromising training speed.

We find that uniform equal-width cuts seldom rival quantile, whereas $k$-means almost always draws level with quantile and occasionally provides a **marked boost**. Across 18 real-world regression datasets, $k$-means *never performs worse* than quantile at the 5% level and cuts MSE by more than half on the most skewed case, though it has a slightly lower MRR (0.65 vs 0.72). When we repeat the sweep with the 63-bin budget recommended for GPU training (Zhang et al., 2017), $k$-means posts five wins and raises its best-case drop to 68% MSE (Table 5), though quantile achieves two small ($\leq 2\%$) wins.

In 15 classification datasets quantile and $k$-means are statistically tied (MRR 0.68 vs 0.7) with gaps $\leq 0.2$ percentage points (pp)–differences that are essentially negligible. Synthetic regression benchmarks tell the same story: when outliers dominate or when bin budget is low, $k$-means often cuts error by more than 20% and can reduce error by as much as 90% in some cases. Taken together, our findings show that $k$-means stays within statistical noise of exhaustive splitting when the bin budget is generous, yet often uncovers critical split points that quantile misses.

To our knowledge, no open-source GBDT package (XGBoost, LightGBM, CatBoost) offers a $k$-means discretizer, and the literature lacks any systematic test of alternative unsupervised binning schemes. We

---

[1]Appendix D benchmarks two other *globally-optimal* discretizers, MILP-optimal (Navas-Palencia, 2022) and 1-D $k$-means (Wang & Song, 2011). These methods either have unreasonable computational requirements or merge features down to $\leq 25$ bins, significantly harming accuracy.

therefore conduct the first large-scale benchmark (33 OpenML datasets plus controlled synthetics) and show that this drop-in option can significantly cut regression MSE in skewed datasets or in low-bin scenarios while adding only a one-off, cacheable preprocessing cost of $\approx 3.5$s per feature to bin 10 M rows on a single Apple M1 thread—making a `bin_method=k-means` flag an immediately actionable upgrade for commercial GBDT libraries.

## 2 Related Work

In recent years, gradient-boosted decision trees (GBDTs) have become the dominant choice for large-scale tabular prediction thanks to highly optimized "histogram" implementations that reduce time complexity from $O(N \log N)$ to $O(N)$ per feature by pre-binning continuous inputs. LightGBM introduced leaf-wise growth together with GPU-friendly, histogram-based training to parallelize histogram-building and split-finding on large datasets (Ke et al., 2017). XGBoost introduces a weighted quantile sketch to obtain approximate split points that are $\epsilon$-accurate even on distributed data (Chen & Guestrin, 2016). CatBoost further refines this pipeline with ordered boosting and efficient handling of categorical features, yielding state-of-the-art accuracy on many public benchmarks (Prokhorenkova et al., 2019). Scikit-learn, a package for easy application of ML algorithms, recently added `HistGradientBoosting` as a re-implementation of these histogram algorithms for GBDTs, providing a convenient baseline for academic comparisons (Pedregosa et al., 2018).

Although the histogram abstraction is now standard, the *binning step itself* has attracted surprisingly little study. All three major libraries—XGBoost, LightGBM, and CatBoost—still build their histograms from equal-frequency (quantile) cuts. Their main differences lie elsewhere: (1) XGBoost replaces the naïve $O(N \log N)$ per-feature sort with a weighted quantile sketch—a streaming algorithm that yields $\epsilon$-accurate percentiles in $O(N)$ time and sub-linear memory, allowing for distributed quantile binning (Chen & Guestrin, 2016); (2) LightGBM applies gradient-based one-side sampling (GOSS) *after* bins are formed, keeping rows with large gradients to speed up tree growth (Ke et al., 2017); (3) CatBoost introduces ordered boosting and specialized handling of categorical features (Prokhorenkova et al., 2019). Thus, despite diverse engineering optimizations, the underlying assumption that equal-frequency bins are "good enough" remains **largely unchallenged.**

Adaptive or data-driven binning has been explored in other settings. Early entropy-based cuts (Fayyad & Irani, 1993) and MDL-guided binning (Dougherty et al., 1995) found that supervised partitions improve decision-tree and Naïve-Bayes accuracy over equal-width binning in classification settings. In federated settings, Ong et al. (2020) begin with equal-mass bins and dynamically merge or split them using gradient entropy to keep communication costs fixed. Our work is largely orthogonal to these settings: we study *library-level* [2], *unsupervised* binning that can serve as the starting point for any histogram-GBDT pipeline, federated or centralized[3]. Although $k$-means is unsupervised, its variance-minimizing objective lets it capture label-relevant tails that equal-mass schemes often smooth over, thus recovering a portion of the gains previously attributed only to supervised or adaptive methods.

In summary, while histogram GBDTs have matured along multiple engineering directions, the underlying assumption that *equal-frequency bins are universally adequate has remained largely unchallenged.* Our work provides the first systematic comparison between quantile, $k$-means and uniform binning on 33 real-world tasks and a controlled synthetic suite, revealing areas where a simple $k$-means swap yields substantial accuracy gains with negligible overhead.

## 3 Theoretical Motivation

Although our contribution is primarily empirical, we provide a short analysis showing that for any *L-Lipschitz* target $f$, $k$-means binning maximizes the tightest possible *lower bound* on the explained variance of $Y$ obtained when treating all values in a given bin as equivalent.

---

[2]By "library-level" we refer to modifications that live inside the GBDT library itself (e.g. LightGBM, XGBoost). The only exposure to the end user is an extra option to specify the `bin_method` – no separate preprocessing script or data-pipeline changes are required.

[3]However, we do conduct a small benchmarking experiment in Appendix D which includes a supervised binning method.

Throughout this section we work in the *population* (distributional) setting: a $K$-binning is a measurable partition $B = \{B_1, B_2, ...B_K\}$ of the feature domain, and we write $\pi_j = \mathbb{P}(X \in B_j)$. Given centers $\{c_1, ..., c_K\}$ the $k$-means quantization (binning) objective is to minimize

$$\sum_{j=1}^{K} \pi_j \mathbb{E}[(X - c_j)^2 | X \in B_j]$$

For any fixed partition $B$, the objective is minimized with respect to the centers when $c_j = \mathbb{E}[X | X \in B_j]$; substituting yields

$$\sum_{j=1}^{K} \pi_j \operatorname{Var}(X | X \in B_j)$$

Because the optimization is taken jointly over all measurable $B$ *and* the centers $c_1, ..., c_K$, population-level $k$-means is therefore *exactly* the problem of minimizing the expected within-bin variance across all $K$ partitions. This mirrors the familiar sample version, where $k$-means minimizes the within-cluster variance of the observed data points (Lloyd, 1982).

**Lemma 1** (Paired–difference identity). *Let $Z$ be a square-integrable random variable and let $Z' \stackrel{d}{=} Z$ be an independent copy of $Z$. Then*

$$\operatorname{Var}(Z) = \frac{1}{2}\mathbb{E}\big[(Z - Z')^2\big] \tag{1}$$

*Proof.* By independence and identical distribution,

$$\mathbb{E}\big[(Z - Z')^2\big] = \tag{2}$$
$$\mathbb{E}[Z^2] + \mathbb{E}[{Z'}^2] - 2\,\mathbb{E}[Z\,Z'] = \tag{3}$$
$$2\,\mathbb{E}[Z^2] - 2\,\big(\mathbb{E}[Z]\big)^2 = \tag{4}$$
$$2\operatorname{Var}(Z) \tag{5}$$

Dividing both sides by 2 yields (1) □

**Theorem 1.** *Let random variables (X, Y) with finite second moments satisfy a $L$-Lipschitz continuous model $Y = f(X) + \epsilon$ with $\mathbb{E}[\epsilon] = 0$, $\epsilon \perp X$, $\operatorname{Var}(\epsilon) = \sigma^2$. Then, k-means binning chooses the bins which maximize the tightest possible lower bound on the explained variance of Y.*

*Proof.* Fixing integer K and measurable binning B as $\{B_1, B_2, ..., B_K\}$, define $\pi_j = \mathbb{P}(X \in B_j)$. We write $\mathbb{E}_j[g_j] = \sum_{j=1}^{K} \pi_j g_j$ and $Var_j(g_j) = \sum_{j=1}^{K} \pi_j \big(g_j - \mathbb{E}_j[g_j]\big)^2$ as the expectation/variance of some function of g across bins under $\{\pi_j\}$.

For regression, we have that the explained variance of a given binning B is

$$\sum_{j=1}^{K} \pi_j \left( \mathbb{E}[Y | X \in B_j] - \mathbb{E}[Y] \right)^2 = \tag{6}$$
$$\operatorname{Var}_j \big( \mathbb{E}[Y | X \in B_j] \big) = \tag{7}$$
$$\operatorname{Var}(Y) - \mathbb{E}_j[\operatorname{Var}(Y | X \in B_j)] \tag{8}$$

By the law of total variance. Since Y and X are fixed before bin choice, the explained variance is maximized when $\mathbb{E}_j[\text{Var}(Y|X \in B_j)]$ is minimized. Using Lemma 1 and the $L$-Lipschitz property of f, we have that

$$\text{Var}(Y|X \in B_j) = \tag{9}$$

$$\frac{1}{2}\mathbb{E}[(Y - Y')^2|X \in B_j] = \tag{10}$$

$$\frac{1}{2}\mathbb{E}[(f(X) - f(X') + \epsilon - \epsilon')^2|X \in B_j] = \tag{11}$$

$$\frac{1}{2}\mathbb{E}[(f(X) - f(X'))^2|X \in B_j] + \sigma^2 \leq \tag{12}$$

$$\frac{L^2}{2}\mathbb{E}[(X - X')^2|X \in B_j] + \sigma^2 = \tag{13}$$

$$L^2 \text{Var}(X|X \in B_j) + \sigma^2 \tag{14}$$

Consequently,

$$\mathbb{E}_j\big[\text{Var}(Y \mid X \in B_j)\big] \leq \mathbb{E}_j\big[L^2 \text{Var}(X \mid X \in B_j) + \sigma^2\big] = L^2\mathbb{E}_j[\text{Var}(X|X \in B_j)] + \sigma^2$$

Since $k$-means minimizes $\mathbb{E}_j[\text{Var}(X|X \in B_j)]$ (as shown earlier), it minimizes this upper bound and therefore maximizes a lower bound on the explained variance of Y. Since this bound is strict when $y = \beta X$ (proof in Appendix A), this is the tightest possible lower bound on the explained variance of Y.

$\square$

Since exact $k$-means binning is impractical for large datasets[4], in Section 4 we replace it with Lloyd's algorithm via `scikit-learn`, which offers a fast heuristic approximation to the exact $k$. We find that this theoretical justification carries over when using Lloyd's algorithm on real-world datasets, with $k$-means binning often obtaining the lowest *realized* error.

## 4 Experiments

We evaluate our discretizers on 33 OpenML tasks and a suite of controlled synthetic benchmarks. An anonymized reproducibility package—source code, logs, and result tables—is provided for reviewers (see the Links section).

### 4.1 Real-world benchmarks

#### 4.1.1 Methodology

**Datasets.** To ensure replicable results we evaluate our models on the OpenML (Vanschoren et al., 2014) benchmark suite described in Grinsztajn et al. (2022) (`study_id` 336 for regression, 337 for binary classification), dropping one task from each track (HIGGS, ZURICH DELAYS) due to computational constraints. The remaining **18 regression** and **15 classification** tasks span $10^3$–$10^6$ instances and 2–420 numeric features. Appendix B lists observations and features for each dataset.

**Binning schemes.** We compare three discretizers: *quantile* (LightGBM default), *uniform* (equal-width), *k-means* (Lloyd with quantile seeding). Unless noted otherwise, all binning schemes use $B = 255$ bins (the common default in LightGBM and scikit-learn, among others).[5]

In all real-world benchmark tables, we also list an "exhaustive search" column, which checks all possible split thresholds without using binning. Because that variant enumerates all possible split thresholds (O(N) per

---

[4]Appendix D presents a brief benchmark of the 1-D dynamic-programming formulation of optimal $k$-means (Wang & Song, 2011). The method's computational cost is so high that it becomes impractical for histogram-based GBDTs, offering no realistic speed–accuracy advantage over Lloyd's algorithm.

[5]Appendix E reports an ablation at the GPU-recommended budget of $B = 63$, as described in Zhang et al. (2017).

node), it is impractically slow on large datasets but serves as a practical upper-bound reference to quantify the accuracy degradation attributable to binning itself.

In Appendix D, we present a brief benchmark of two further binning schemes and show that neither achieves an attractive speed–accuracy trade-off.

**Learners and tuning.** Commercial-grade libraries such as LightGBM, XGBoost, and CatBoost apply additional preprocessing layers—e.g. LightGBM's exclusive-feature bundling, or XGBoost's weighted quantile sketch (Ke et al., 2017) (Chen & Guestrin, 2016). These steps are re-executed even when the input has already been pre-binned to 255 distinct values, producing a second, library-specific histogram that blurs the effect we wish to measure. To isolate the contribution of the *external* discretizer itself, we therefore run all main-paper experiments with scikit-learn's vanilla `GradientBoostingRegressor/Classifier`, where the model consumes our bins exactly as supplied. In a supplementary run with XGBoost (Appendix F)—configured to bypass its internal quantile sketch—we observed similar results on representative regression datasets, suggesting that the choice of baseline model does not influence our conclusions.

For every *(dataset, binning, learner)* triple we run a 30-trial `RandomizedSearchCV` with 5-fold CV on hyper-parameters shown in Appendix C. Each experiment is repeated over **20 random train/test splits** (80/20) to estimate variability.

**Compute resources.** All real-world experiments were conducted on an academic slurm cluster, on 48-core Intel Xeon Platinum 8268 CPUs @ 2.90 GHz. The experiments ran in 120h wall-clock, doing 1792 core-hours of work, and peaking at 7.7GB memory. Additional exploratory runs on the same cluster amounted to no more than 10k CPU-hours ($\approx 6 \times$ the final sweep), as confirmed from Slurm accounting over the project period.

**Metrics and statistics.** We report mean squared error (MSE) for regression and ROC-AUC for binary classification. Regression MSE values span several orders of magnitude, so each row of our results table presents them in scientific notation with the common exponent factored out on the *left* of the dataset name (e.g. "Brazilian Houses $(10^{-3})$"). This keeps the numeric columns directly comparable across datasets.

To provide a scale-free summary metric, we compute a macro-averaged mean-reciprocal-rank (MRR, $\uparrow$ is better). For each dataset the three histogram methods (exhaustive search excluded) are ranked; the reciprocal rank is then averaged over datasets, with each dataset contributing one vote.

Paired two-sided t-tests (n=20 splits) compare the top-ranked discretizer with the runner-up (exhaustive search excluded) on each dataset. All resulting p-values are jointly adjusted with the Benjamini–Hochberg procedure (Benjamini & Hochberg, 1995); cells that stay significant at the $\alpha = 0.05$ level after this correction are bolded.

In Appendix H we mirror all experimental tables – including those in Appendices E and F – and add the standard error across random seeds.

### 4.1.2 Results

Tables 2a and 2b report mean test-set scores averaged over 20 random 80:20 splits, with per-dataset winners flagged by paired t-tests, as described in section 4.1.1.

- **Regression (18 tasks)** Although $k$-means attains a slightly lower MRR compared to quantile binning (0.65 vs. 0.72), it is never statistically outperformed by either quantile or uniform binning at the 5% significance level across all evaluated datasets. However, $k$-means wins outright in 3 datasets, most strikingly by 55%, on the Brazilian Houses dataset. Its other wins range from 1-2%. Brazilian Houses is especially unique, because it contains the most highly skewed input dataset (averaged by column) among those in the benchmark. In addition, the extreme predictor values (e.g. high property tax) coincide with extreme target values (rent price), so isolating these points is crucial. Across the full regression suite, $k$-means performs $\geq 0.3\%$ worse than the exhaustive split method on just three datasets; of those, quantile performs statistically significantly worse than

$k$-means on two (and significantly better on none). This suggests that $k$-means retains almost all of the predictive value of exhaustive splitting when extra cuts add little value, yet still recovers key tail-driven split-points that quantile overlooks. In section 4.2, we conducted in-depth analysis into this relationship between $k$-means performance and highly-skewed data.

Zhang et al. (2017) recommend using a budget of $B = 63$ when training GBDTs on a GPU; in Appendix E, we rerun the regression suite to analyze this scenario. Under this tighter budget, $k$-means now holds an 8% lead on CPU ACT and widens its lead on BRAZILIAN HOUSES to 66%, while quantile gains a statistically significant 2.5% lead on SUPERCONDUCT and a 0.5% lead on DIAMONDS. In Appendix E, we hypothesize two mechanisms for quantile's small wins in the low-bin regime: (1) k-means may spend a larger *proportion* of a small bin budget isolating extremes that may not be label-relevant, and (2) coarser histograms also introduce greater variance, so edge-label alignment can occur by chance.

- **Classification (15 tasks).** Quantile and $k$-means achieve virtually identical MRR (0.68 vs. 0.70). Quantile attains statistically significant wins on two datasets (5% level), while $k$-means claims none; however, even in those two cases the advantage is at most 0.2 pp—practically negligible.

  Notably, the effect of $k$-means is much smaller for classification than it is for regression. The core difference is that baseline classification losses are bounded, whereas squared-error in regression grows without limit. When many distinct, widely spaced numeric values fall into the same bin, a tree must assign them a single prediction. In regression that prediction is their mean, so any extreme value in the bin incurs a squared error that grows with its distance from the mean; a few such outliers can dominate the total MSE. In binary classification, collapsing diverse values has a softer effect: the tree assigns the bin a single class-probability. For a given bin, the worst accuracy it can give is 0.5 (predicting majority class for all), and the worst log-loss is $\ln 2$ (predicting 0.5 for all) - both fixed, finite penalties that do not explode with feature magnitude. Thus isolating rare, label-relevant extreme points with $k$-means dramatically lowers regression MSE but nudges classification metrics by only a few hundredths.

We also pre-binned three representative regression datasets with either quantile or our $k$-means cuts (B = 255) and trained XGBoost using its exact (i.e. exhaustive) tree builder. As Appendix F shows, $k$-means retains existing ties while reducing MSE on BRAZILIAN HOUSES by 64%, confirming the lift translates unchanged to a mainstream GBDT engine when using exhaustive no-histogram methods.

These patterns–large gains when label-relevant tails exist or when bin budget is low–suggest *histogram resolution* and *outlier mass* are the controlling factors. We investigate those mechanisms systematically via the synthetic suite in section 4.2.

## 4.2 Synthetic diagnostics

We complement the real-world benchmarks with a controlled suite of five synthetic studies that probe exactly *when $k$-means binning beats equal-frequency cuts*. All runs share the same generator (Alg. 1) which computes a target as the sum of 3 columns combined with Gaussian noise and allows independent control of *outlier mass*, *outlier magnitude*, *multi-modality*, *bin budget B*, and *sample size n*. We split the dataset into 80/20 train-test and run scikit-learn's `GradientBoostingRegressor`. We keep the scikit-learn defaults (100 trees, learning-rate = 0.01) for consistency, and simply raise the depth to 5 and set subsample = 0.8 to give the model adequate capacity and standard stochastic regularization without masking the binning effects.

Each cell in Figs. 1–2 reports the mean relative MSE reduction

$$\Delta\% = 100 \times \frac{\text{MSE}_{\text{quantile}} - \text{MSE}_{k\text{-means}}}{\text{MSE}_{\text{quantile}}}$$

over **50 i.i.d. dataset/split draws** (stars mark cells where the two methods perform statistically significantly different at the 95% confidence level, after performing a Benjamini–Hochberg false-discovery-rate correction). Quantile statistically ties or loses to $k$-means across all of our experiments, with losses as high as **90%** in certain cases.

| Dataset Name | Quantile | Uniform | $k$-means | Exhaustive |
|---|---|---|---|---|
| **Regression (MSE)** | | | | |
| cpu_act $(10^0)$ | 5.043 | 5.094 | 4.965 | 5.092 |
| pol $(10^1)$ | 3.337 | 3.337 | 3.337 | 3.347 |
| elevators $(10^{-6})$ | 4.861 | 4.842 | 4.862 | 4.881 |
| wine_quality $(10^{-1})$ | 4.096 | 4.089 | 4.128 | 4.117 |
| Ailerons $(10^{-8})$ | 2.518 | 2.526 | 2.523 | 2.519 |
| houses $(10^{-2})$ | 5.274 | 5.358 | 5.283 | 5.292 |
| house_16H $(10^{-1})$ | 3.338 | 3.674 | 3.415 | 3.262 |
| diamonds $(10^{-2})$ | 5.458 | 5.602 | 5.464 | 5.459 |
| Brazilian_houses $(10^{-3})$ | 5.382 | 20.272 | **2.433** | 2.156 |
| Bike_Sharing_Demand $(10^3)$ | 9.697 | 9.697 | 9.697 | 9.694 |
| nyc-taxi-green-dec-2016 $(10^{-1})$ | 1.553 | 1.959 | **1.522** | 1.320 |
| house_sales $(10^{-2})$ | 3.183 | 3.217 | 3.175 | 3.206 |
| sulfur $(10^{-4})$ | 4.649 | 4.698 | 4.663 | 4.779 |
| medical_charges $(10^{-3})$ | 6.686 | 7.153 | **6.600** | 6.584 |
| MiamiHousing2016 $(10^{-2})$ | 2.259 | 2.291 | 2.264 | 2.309 |
| superconduct $(10^2)$ | 1.012 | 1.036 | 1.015 | 1.038 |
| yprop_4_1 $(10^{-4})$ | 9.457 | 9.481 | 9.464 | 9.474 |
| abalone $(10^0)$ | 4.775 | 4.788 | 4.767 | 4.759 |
| **Regression MRR** | 0.72 | 0.43 | 0.65 | |
| **Classification (ROC AUC)** | | | | |
| credit | 0.857 | 0.827 | 0.857 | 0.857 |
| electricity | **0.950** | 0.918 | 0.948 | 0.960 |
| covertype | **0.933** | 0.931 | 0.932 | 0.931 |
| pol | 0.999 | 0.999 | 0.999 | 0.999 |
| house_16H | 0.951 | 0.947 | 0.951 | 0.950 |
| MagicTelescope | 0.931 | 0.931 | 0.931 | 0.930 |
| bank-marketing | 0.886 | 0.886 | 0.886 | 0.886 |
| MiniBooNE | 0.983 | 0.967 | 0.983 | 0.982 |
| eye_movements | 0.705 | 0.709 | 0.709 | 0.718 |
| Diabetes130US | 0.647 | 0.647 | 0.647 | 0.647 |
| jannis | 0.868 | 0.867 | 0.868 | 0.867 |
| default-of-credit-card-clients | 0.781 | 0.779 | 0.781 | 0.780 |
| Bioresponse | 0.861 | 0.859 | 0.860 | 0.858 |
| california | 0.967 | 0.962 | 0.967 | 0.966 |
| heloc | 0.798 | 0.798 | 0.798 | 0.798 |
| **Classification MRR** | 0.68 | 0.41 | 0.70 | |

Table 1: Summary of quantile, uniform, and $k$-means binning on 33 real-world regression and classification datasets.

**Compute Resources.** All synthetic experiments were executed on a MacBook Pro (Apple M1 Pro, 8 threads, 16 GB RAM, macOS 15.0). They completed in about 1.55 hours of wall-clock time, used roughly 4.6 core-hours in total, and never exceeded 0.5 GB of memory. Preliminary experiments roughly tripled this compute cost. These compute costs are essentially negligible in comparison to the much larger compute requirements of testing on real-world benchmarks.

**(1) Varying outlier mass and magnitude.** Figure 1a fixes a single Gaussian mode and varies the fraction of outliers (0–5%) against their scale $\beta$, drawing each outlier from an exponential tail with scale parameter $\beta \in \{5, 10, 15, 20\}$. With just 1% outliers, $k$-means outperforms quantile by more than 50% regardless of outlier scale, though the gain increases with scale (at $20\sigma$, the gain is **75%**). Gains slightly plateau as the tail fraction nears 5%, suggesting that the most extreme advantages stem from a smaller, label-relevant tail–as "outliers" fill a greater portion of the dataset, quantile tends to isolate them more effectively, though still not as well as $k$-means does.

**(2,3) Multi-modality.** Figure 1b fixes the outlier mass at 1% and varies the number of density modes against the outlier scale $\beta$. Although we expected $k$-means to perform better on multi-modal data, by isolating each mode, we instead see small, if any, improvements as the number of modes increase. This suggests that isolating outliers is more important to a GBDT's performance than isolating modes, even when modes are well-separated. Regardless, when outliers exist, the $k$-means gain remains large (40-80%) for all mode counts, rising with $\beta$; even with 10 well-separated peaks a $20\sigma$ tail still yields 70%+ error reduction.

Figure 1c keeps the tail magnitude fixed ($\beta = 5$) and increases both the mode count and the outlier fraction. We see a similar pattern here as in experiment (1): as outliers constitute a larger fraction of the dataset, the $k$-means gap decreases as quantile already allocates bins to the larger tail. At the bottom row, when there are no outliers, we see confirmation that with $B = 255$, quantile isolates tails of multiple normal distributions as effectively as $k$-means does.

Together, experiments (2)–(3) show that multi-modality has little effect on the $k$-means gap, neither providing a benefit to $k$-means nor diluting existing gaps in the presence of outliers.

**(4) Sample size.** Figure 2a varies the effective histogram resolution by increasing the sample size while keeping $B = 255$. With no outliers, the two methods perform within 3%. However, with even 1% outliers, bin budget comes into play; when each bin contains 64-128 observations, the data are quantized coarsely and $k$-means cuts error by nearly **90%**. This gap collapses (though, still exists) as the binner quantizes data more smoothly.

**(5) Bin Budget $B$.** Conversely, Fig. 2b fixes the sample size while varying the number of bins. Here, the binning methods differ even without outliers; $k$-means outperforms quantile on a pure normally-distributed dataset by **43%** when using only 16 bins. With significant outliers, we see similar results as in experiment (4); $k$-means outperforms quantile by nearly **90%** when outliers constitute 1-5% with 16-32 bins. The top row corresponds to a data set drawn entirely from an exponential distribution. Despite its heavy right-hand tail, it behaves similar to the pure-Gaussian case—especially once the bin budget exceeds about 64—indicating that $k$-means' advantage is triggered by the *presence of a small, label-relevant tail alongside a dense bulk*, not by heavy-tailedness per se.

**Key take-aways from the synthetic suite.**

1. **Safe default.** Even in the pure-Gaussian settings (0% outliers) $k$-means never statistically under-performs and sometimes improves MSE by 2–3%, thanks to slightly finer cuts in the sparse tails that quantile assigns a single bin.

2. **Huge upside when a small tail drives the target.** With only 1% of samples lying $10$–$20\sigma$ above the bulk, $k$-means cuts error by 50–90% (experiments 1–3), and the gain remains large even when that tail co-exists with up to twenty well-separated modes.

3. **Most valuable at coarse histograms.** Even in low-skew datasets, $k$-means can outperform quantile when the bin budget is tight (32-64 bins or $\sim$ 80-120 observations per bin in experiments 4-5); as cells hold fewer samples, the two schemes converge. Thus $k$-means binning is especially useful whenever memory or pass constraints force a low-resolution histogram.

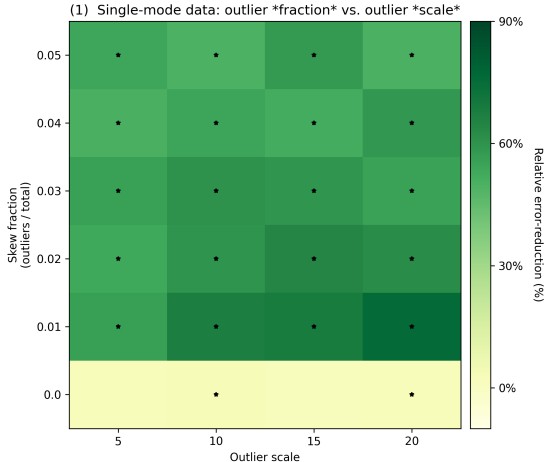

(a) (1) Single-mode data: outlier *scale* $\beta$ (cols) vs. outlier *fraction* (rows)

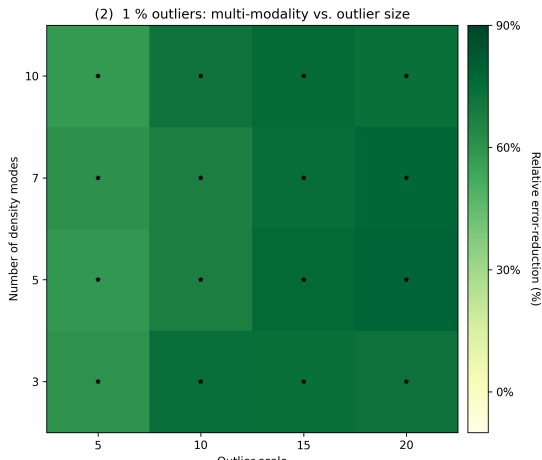

(b) (2) Fixed outlier freq. 1%; outlier scale $\beta$ (cols) vs. multi-modality (rows)

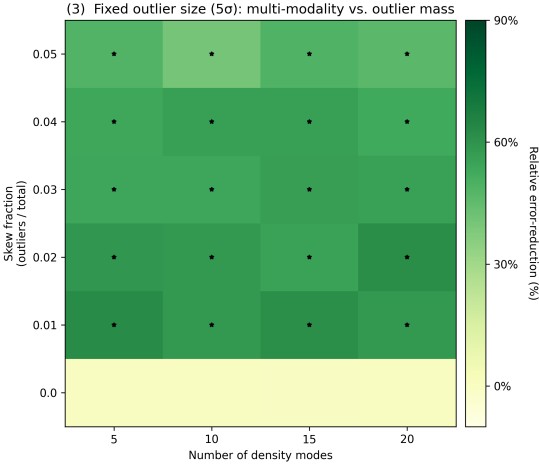

(c) (3) Fixed outlier scale $\beta = 5\sigma$; outlier freq. (cols) vs. multi-modality (rows)

Figure 1: Synthetic experiments 1–3: relative MSE reduction ($\Delta\%$) of $k$-means over quantile binning. Each cell averages 50 runs; greener is better.

## 5 Computational Efficiency

Before training a model, GBDT packages that utilize histogram-binning discretize each continuous feature into a small fixed budget of bins ($B = 255$ as default for CPU training in many packages). Once the edges are set, tree growing scans *bins* rather than individual samples, so the cost of computing split gains falls from $O(n)$ to $O(B)$ for *every* discretizer—quantile, equal-width, or our $k$-means. The model training complexity is therefore identical; the only extra work is the one-off bin-construction pass. Because discretization happens before model training, libraries can offer caching of bin edges if users plan to run multiple models on the same training set – for example, if they are conducting a hyperparameter search.

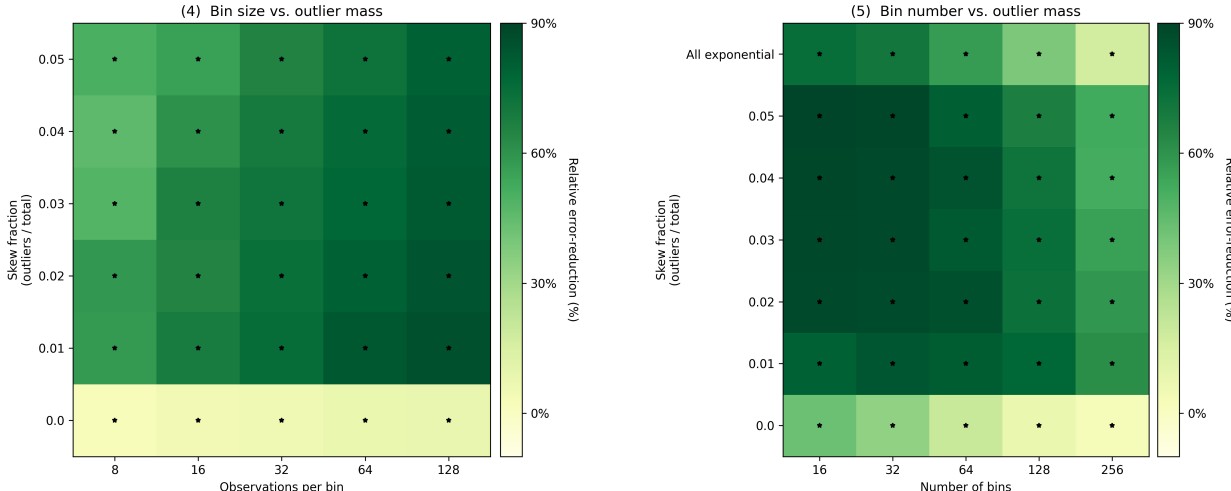

Figure 2: Synthetic experiments 4–5: effect of histogram resolution. Left: variable sample size at fixed bin budget. Right: variable bin budget at fixed sample size.

Figure 3 benchmarks the bin-construction pass on a single-thread Apple M1 for uniformly distributed data ranging from $10^4$ to $10^7$ rows. Times are per feature; to estimate wall-clock cost for $d$ columns, multiply the reported value by $d$, as both 1-D binning methods and GBDTs scale linearly in $d$. Equal-width and quantile curves remain close across the range, suggesting essentially equivalent big-O time complexity and constant factor. $k$-means with quantile seeding follows the same slope but with a higher intercept: on ten million rows quantile finishes in 2s, whereas $k$-means requires 5.5s; at one million rows the gap is a mere 0.3s. We also experimented with $k$-means++ seeding (Arthur & Vassilvitskii, 2007), but saw no meaningful accuracy improvement over $k$-means with quantile seeding. Since the extra $O(nk)$ initialization pass in $k$-means++ made preprocessing substantially slower, we decided to use quantile seeding throughout.

We note that quantile, uniform, and quantile-seeded $k$-means consume less than one-third of total training time. This will remain the case regardless of the number of features $d$, since both binning and GBDT training itself scale linearly in $d$. In addition, as the cost to bin is incurred once and can be cached, the extra $\approx 3.5$ seconds per feature on 10M+ row datasets are negligible in practice—especially relative to the strong error reduction we observe on skewed regression tasks. For datasets that exceed available RAM, a streaming 1-D $k$-means variant (Sculley, 2010) can be substituted without changing the API; we leave evaluating this option on truly massive tables to future work.

## 6 Conclusion

This work revisited a seemingly innocuous design choice that underlies all modern histogram-based GBDT implementations: *how should one place the bin boundaries?* Our systematic study across **33 real-world tasks**, a five-factor synthetic suite, and a new lower-bound theorem, shows that the long-standing default of equal-frequency cuts is not sacrosanct.

We find that, in datasets with *small, label-relevant tails*, or in scenarios with low bin budgets, $k$-means can cut error significantly, sometimes by *more than half*. In particular, we find that $k$-means matches exhaustive splitting in benign cases and consistently outperforms quantile whenever tails or tight bin budgets make histograms too coarse.

Because training time after binning is identical for all binning methods, the only extra cost is a one-off, cacheable preprocessing pass—essentially negligible at roughly 3.5s for 10M rows on a single Apple M1 thread.

In short, today's "one-size-fits-all" quantile cuts leave easy accuracy on the table. A single new option—`bin_method=k-means`—would cost nothing at training time, remain fully backward-compatible, and

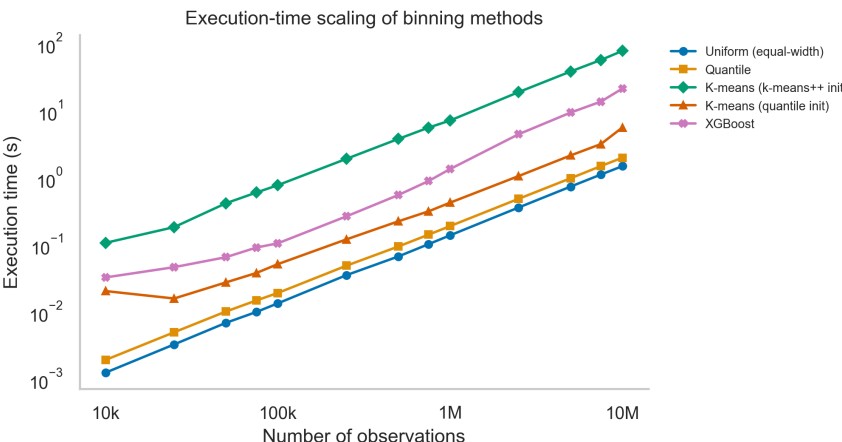

Figure 3: Wall-clock time to bin/train a model on a single continuous feature on a single M1 CPU thread (log–log scale).

yield double-digit accuracy gains in common scenarios with skewed targets or tight bin budgets, such as the 32–64-bin GPU setting.

We therefore recommend that mainstream GBDT libraries expose a `bin_method=k-means` option and adopt it as the default whenever the bin budget is constrained, as on GPUs.

# 7 Limitations and Future Work

We deliberately ran our study with scikit-learn's "vanilla" histogram GBDT, eliminating all library-specific preprocessing. A small replica (found in Appendix F) using XGBoost's exact method shows the same pattern, but a full integration into production pipelines—e.g. LightGBM's exclusive-feature bundling, XGBoost's quantile sketch, CatBoost's ordered targets—remains an engineering task for future work. Our results therefore speak most directly to library maintainers, who can offer a `bin_method=k-means` flag once these code paths are updated.

As shown throughout, $k$-means outperforms quantile most when datasets have a *small, label-relevant tail or when bin budget is tight*. If the tail information can already be reconstructed from other features, or if the target changes sharply at unknown boundaries inside a dense bulk (e.g. a normally distributed predictor with a step change at $x = \mu$), the advantage shrinks considerably. Likewise, if users have access to large bin budgets, they should expect to see smaller gaps between $k$-means and quantile binning.

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

## A  Proof of Strictness for Linear Model

We begin the proof as in Section 3, with the additional assumption that $Y = \beta X + \epsilon$. In particular, the proof remains the same up until (12), wherein it continues as follows:

$$\frac{1}{2}\mathbb{E}[(f(X) - f(X)')^2 | X \in B_j] + \sigma^2 = \tag{15}$$

$$\frac{1}{2}\mathbb{E}[(\beta X - \beta X')^2 | X \in B_j] + \sigma^2 = \tag{16}$$

$$\beta^2 \operatorname{Var}(X | X \in B_j) + \sigma^2 \tag{17}$$

Consequently,

$$\mathbb{E}_j\big[\operatorname{Var}(Y \mid X \in B_j)\big] = \mathbb{E}_j\big[\beta^2 \operatorname{Var}(X \mid X \in B_j) + \sigma^2\big] = \beta^2 \mathbb{E}_j[\operatorname{Var}(X | X \in B_j)] + \sigma^2$$

Since k-means binning minimizes $\mathbb{E}_j[\operatorname{Var}(X | X \in B_j)]$, it also maximizes the explained variance of Y obtained when treating all values in a given bin as equivalent.

## B  Dataset Description

The following table describes observations and features from each dataset. For a given dataset, we calculate the skew by averaging the skew of each of the X columns. These datasets range from $10^3 - 10^6$ observations, and up to 400+ features. We calculate skew using the default `scipy` Fisher–Pearson sample skewness (`scipy.stats.skew`, `bias=True`). Recomputing with the unbiased variant (`bias=False`) changes values by $< 0.005$, leaving all printed numbers unchanged.

| Dataset | Skew | #Obs | #Feat |
|---|---|---|---|
| cpu_act | 5.40 | 8 192 | 22 |
| pol | 4.78 | 15 000 | 27 |
| elevators | -0.60 | 16 599 | 17 |
| wine_quality | 1.36 | 6 497 | 12 |
| Ailerons | 0.60 | 13 750 | 34 |
| houses | 2.23 | 20 640 | 9 |
| house_16H | 6.14 | 22 784 | 17 |
| diamonds | 1.03 | 53 940 | 7 |
| Brazilian_houses | 30.23 | 10 692 | 9 |
| Bike_Sharing_Demand | 0.06 | 17 379 | 7 |
| nyc-taxi-green-2016 | 2.56 | 581 835 | 10 |
| house_sales | 2.40 | 21 613 | 16 |
| sulfur | -0.03 | 10 081 | 7 |
| medical_charges | 5.02 | 163 065 | 4 |
| MiamiHousing2016 | 0.93 | 13 932 | 14 |
| superconduct | 0.67 | 21 263 | 80 |
| yprop_4_1 | -0.29 | 8 885 | 43 |
| abalone | 0.62 | 4 177 | 8 |
| **zurich_transport** | 1.00 | 5 465 575 | 9 |

(a) Regression datasets

| Dataset | Skew | #Obs | #Feat |
|---|---|---|---|
| credit | 20.17 | 16 714 | 11 |
| electricity | 12.07 | 38 474 | 8 |
| covertype | 0.28 | 566 602 | 11 |
| pol | 5.34 | 10 082 | 27 |
| house_16H | 5.78 | 13 488 | 17 |
| MagicTelescope | 0.58 | 13 376 | 11 |
| bank-marketing | 3.65 | 10 578 | 8 |
| MiniBooNE | -11.68 | 72 998 | 51 |
| **Higgs** | 1.56 | 940 160 | 25 |
| eye_movements | 3.18 | 7 608 | 21 |
| Diabetes130US | 5.48 | 71 090 | 8 |
| jannis | -0.22 | 57 580 | 55 |
| default-credit-card | 5.55 | 13 272 | 21 |
| Bioresponse | 6.95 | 3 434 | 420 |
| california | 19.56 | 20 634 | 9 |
| heloc | -0.32 | 10 000 | 23 |

(b) Classification datasets

Table 2: Dataset characteristics. Boldface marks the datasets excluded from the final benchmark due to computational limits.

## C  Hyper-parameter Search Space

Before tuning, we draw each trial's configuration from the distributions in Table 3. Thirty trials give a good trade-off between search cost and performance.

Table 3: Randomized search space for `GradientBoostingRegressor`. Each of the 30 trials in `RandomizedSearchCV` draws one value from every distribution. Distributions follow `scipy.stats` (Virtanen et al., 2020) notation: `uniform(loc, scale)` samples from $[loc, loc + scale]$.

| Hyper-parameter | Distribution / range |
|---|---|
| # Estimators | randint(20, 300) |
| Learning rate | loguniform($10^{-3}$, 0.5) |
| Max depth | randint(3, 6) |
| Subsample | uniform(0.5, 0.5) |
| Max features | uniform(0.5, 0.5) |

## D  Small Benchmark for Alternative Binners

Although the body of the paper focuses on *quantile*, *uniform* and *Lloyd–k-means*, we ran a miniature benchmark to gauge the price of more *optimal* schemes. In particular, we compared

- **MILP–optimal** regression binning (Navas-Palencia, 2022),
- **1-D $k$-means** solved by dynamic programming (Wang & Song, 2011), and
- the two fast baselines (quantile and Lloyd–$k$-means).

Given the extended training time of 1-D $k$-means, we restricted the test to a small (ABALONE, $4\,177 \times 8$) and a medium (DIAMONDS, $53\,940 \times 7$) dataset. All runs use the hyper-parameter grid in Appendix C. Binning time is reported *per split*; the full $30 \times 5$ CV grid therefore costs $150\times$ the numbers shown.

| Binning Method | **Abalone** (0.22s to train) | | **Diamonds** (1.82s to train) | |
|---|---|---|---|---|
| | *Binning time (s)* | *MSE* ($10^0$) | *Binning time (s)* | *MSE* ($10^{-2}$) |
| Quantile | 0.003 | 4.76 | 0.03 | 5.30 |
| $k$-means | 0.028 | 4.65 | 0.39 | 5.36 |
| MILP | 0.161 | 5.42 | 0.19 | 8.80 |
| 1-D $k$-means optimal | 0.559 | 4.73 | 6.75 | 5.31 |

Table 4: Wall-clock preprocessing time (single M1 thread) and test MSE for two representative datasets. Average time to train GBDT on pre-binned data is listed next to each dataset's name.

**Take-away.**

MILP-optimal runs in a reasonable time but often merges features down to $\leq 25$ bins, hurting accuracy; DP $k$-means matches Lloyd on error but is >15x slower (and >2x slower than training itself). Therefore, neither option is competitive for large-scale histogram GBDT training.

## E  Real-world Benchmark with Low Bin Budget

To understand the effect of bin budget on real-world datasets, we conducted equivalent experiments as described in section 4.1 with $B = 63$. In LightGBM's GPU implementation, Zhang et al. (2017, p.7, p.13) note that using fewer bins (6-bit packing with one reserved value, i.e., 63 usable bins) can increase available per-workgroup workload and reduce local memory requirements, both of which speed up training. They

report significant speedups at this setting, so we rerun our real-world regression suite with B=63 to reflect the GPU-relevant regime and report those results below.

Note specifically the new gaps in the CPU ACT and POL datasets (10% and 2.7% respectively), and the even larger gap on BRAZILIAN HOUSES (68%). Quantile now wins on two datasets, by $< 3\%$. We hypothesize two mechanisms for quantile's two small wins in the low-bin regime. First, k-means can allocate the same *number* of bins to extreme values as at higher B, but when B is small those bins consume a much larger *proportion* of the total budget – which can sometimes be counterproductive if the tails are not label-relevant. Second, quantile binning may simply win by chance – in low-bin regimes, model performance becomes sensitive to dataset quirks, so accidental alignments between bin edges and label structure can drive small empirical wins.

Table 5: Summary of quantile, uniform, and $k$-means binning on 18 real-world regression datasets, with bin budget $B = 63$. Bolded values imply statistical significance over the second-best method at p = 0.05, after application of Benjamini–Hochberg across all real-world benchmarks.

| Dataset Name | Quantile | Uniform | $k$-means | Exhaustive |
|---|---|---|---|---|
| **Regression (MSE)** | | | | |
| cpu_act $(10^0)$ | 5.579 | 5.416 | **5.140** | 5.092 |
| pol $(10^1)$ | 3.483 | 3.556 | 3.421 | 3.347 |
| elevators $(10^{-6})$ | 4.936 | 4.927 | **4.859** | 4.881 |
| wine_quality $(10^{-1})$ | 4.142 | 4.168 | 4.114 | 4.117 |
| Ailerons $(10^{-8})$ | 2.529 | 2.525 | 2.510 | 2.519 |
| houses $(10^{-2})$ | 5.451 | 6.030 | **5.409** | 5.292 |
| house_16H $(10^{-1})$ | 3.419 | 3.881 | 3.398 | 3.262 |
| diamonds $(10^{-2})$ | **5.501** | 5.810 | 5.526 | 5.459 |
| Brazilian_houses $(10^{-3})$ | 6.507 | 34.465 | **2.222** | 2.156 |
| Bike_Sharing_Demand $(10^3)$ | 9.694 | 9.693 | 9.675 | 9.694 |
| nyc-taxi-green-dec-2016 $(10^{-1})$ | 1.767 | 2.268 | 1.763 | 1.320 |
| house_sales $(10^{-2})$ | 3.195 | 3.322 | 3.189 | 3.206 |
| sulfur $(10^{-4})$ | 5.089 | 4.627 | 4.589 | 4.779 |
| medical_charges $(10^{-3})$ | 7.365 | 13.980 | **6.737** | 6.584 |
| MiamiHousing2016 $(10^{-2})$ | 2.267 | 2.312 | 2.289 | 2.309 |
| superconduct $(10^1)$ | **9.931** | 10.825 | 10.193 | 10.383 |
| yprop_4_1 $(10^{-4})$ | 9.425 | 9.432 | 9.435 | 9.474 |
| abalone $(10^0)$ | 4.775 | 4.837 | 4.790 | 4.759 |
| **Regression MRR** | 0.59 | 0.39 | 0.85 | |

## F   Re-running Representative Datasets on XGBoost

To confirm that our experiments carry over to commercial GBDT algorithms, we compared our $k$-means binner with quantile and uniform on three representative regression datasets. We find that $k$-means achieves a greater win on BRAZILIAN HOUSES, while remaining statistically indistinguishable from quantile ($\alpha = 0.05$) on CPU ACT and SUPERCONDUCT.

Table 6: Summary of binning methods ($B = 255$) on three real-world representative regression datasets using XGBoost's exact (exhaustive) method. Bolded values imply statistical significance over the second-best method at p = 0.05, after application of Benjamini–Hochberg across all real-world benchmarks.

| Dataset Name | Quantile | Uniform | $k$-means | Exhaustive |
|---|---|---|---|---|
| **Regression (MSE)** | | | | |
| cpu_act ($10^0$) | 5.047 | 5.204 | 5.454 | 5.631 |
| Brazilian_houses ($10^{-3}$) | 4.555 | 19.303 | **1.655** | 1.639 |
| superconduct ($10^2$) | 1.003 | 1.038 | 1.014 | 1.032 |

## G    Synthetic-Data Generator

Algorithm 1 shows the procedure used to create the five diagnostic suites discussed in Section 4.2. We motivated this algorithm as a combination of the simplest building blocks that yield the distributional features we wanted to probe (multi-modality and label-relevant outliers).

First, a Gaussian mixture with `n_modes` components creates a controllable number of modes. We fix `dist=4` so that adjacent modes share $\approx 2.5\%$ of their mass – enough to avoid empty bins while keeping the peaks distinct.

Second, we standardize each feature to equalize scale before any outliers are introduced.

Third, an exponential distribution of size $\beta$ applied to a `p_out` fraction of samples generates controllable outlier magnitude and frequency.

Finally, a simple linear target ($y_i = \sum_j X_{ij} + \epsilon_i$) ensures the exponential tails are label-relevant (i.e. extreme $X$ maps to extreme $Y$).

---

**Algorithm 1** MakeSynth($n_{\text{obs}}, n_{\text{feat}}, n_{\text{modes}}, \text{dist}, p_{\text{out}}, \beta$)

---

1: ▷ *Generate input matrix* **X**
2: **for** $j \leftarrow 1$ to $n_{\text{feat}}$ **do**
3:     **for** $i \leftarrow 1$ to $n_{\text{obs}}$ **do**
4:         Draw mode index $m_i \sim \text{Uniform}\{1, \ldots, n_{\text{modes}}\}$.
5:         Set mean $\mu_{m_i}$ from linspace($0, \text{dist} * (n_{\text{modes}} - 1), n_{\text{modes}}$).
6:         $X_{ij} \sim \mathcal{N}(\mu_{m_i}, 1)$.
7:     **end for**
8:     ▷ *Standardize features and inject outliers*
9:     $\mu_j \leftarrow \text{mean}(X_{:j})$
10:    $\sigma_j \leftarrow \text{std}(X_{:j})$
11:    **for** $i \leftarrow 1$ to $n_{\text{obs}}$ **do**
12:        $X_{ij} \leftarrow \frac{X_{ij} - \mu_j}{\sigma_j}$
13:        With prob. $p_{\text{out}}$ replace $X_{ij} \leftarrow X_{ij} + \text{Exp}(\beta)$
14:    **end for**
15: **end for**
16: ▷ *Generate linear target with small noise*
17: $y_i \leftarrow \sum_{j=1}^{n_{\text{feat}}} X_{ij} + \varepsilon_i, \ \varepsilon_i \sim \mathcal{N}(0, 0.1^2)$
18: **return** (**X**, **y**)

---

## H    Variability of Experiments

Here we provide copies of all previous experimental tables in the main text as well as in the appendix, with standard errors of the mean across random states included. As before, **bolded** values imply statistical significance over the second-best method at p = 0.05, after application of Benjamini–Hochberg.

### H.1 255-bin results

| Dataset Name | Quantile | Uniform | $k$-means | Exhaustive |
|---|---|---|---|---|
| **Regression (MSE)** | | | | |
| cpu_act $(10^0)$ | 5.043 (0.170) | 5.094 (0.105) | 4.965 (0.107) | 5.092 (0.122) |
| pol $(10^1)$ | 3.337 (0.086) | 3.337 (0.086) | 3.337 (0.086) | 3.347 (0.088) |
| elevators $(10^{-6})$ | 4.861 (0.067) | 4.842 (0.063) | 4.862 (0.068) | 4.881 (0.068) |
| wine_quality $(10^{-1})$ | 4.096 (0.050) | 4.089 (0.046) | 4.128 (0.054) | 4.117 (0.053) |
| Ailerons $(10^{-8})$ | 2.518 (0.022) | 2.526 (0.021) | 2.523 (0.020) | 2.519 (0.021) |
| houses $(10^{-2})$ | 5.274 (0.047) | 5.358 (0.046) | 5.283 (0.046) | 5.292 (0.040) |
| house_16H $(10^{-1})$ | 3.338 (0.111) | 3.674 (0.133) | 3.415 (0.136) | 3.262 (0.132) |
| diamonds $(10^{-2})$ | 5.458 (0.020) | 5.602 (0.021) | 5.464 (0.022) | 5.459 (0.022) |
| Brazilian_houses $(10^{-3})$ | 5.382 (1.087) | 20.272 (0.520) | **2.433** (0.625) | 2.156 (0.576) |
| Bike_Sharing_Demand $(10^3)$ | 9.697 (0.041) | 9.697 (0.041) | 9.697 (0.041) | 9.694 (0.042) |
| nyc-taxi-green-dec-2016 $(10^{-1})$ | 1.553 (0.011) | 1.959 (0.008) | **1.522** (0.010) | 1.320 (0.016) |
| house_sales $(10^{-2})$ | 3.183 (0.021) | 3.217 (0.022) | 3.175 (0.022) | 3.206 (0.020) |
| sulfur $(10^{-4})$ | 4.649 (0.361) | 4.698 (0.407) | 4.663 (0.358) | 4.779 (0.383) |
| medical_charges $(10^{-3})$ | 6.686 (0.035) | 7.153 (0.042) | **6.600** (0.036) | 6.584 (0.035) |
| MiamiHousing2016 $(10^{-2})$ | 2.259 (0.032) | 2.291 (0.034) | 2.264 (0.032) | 2.309 (0.031) |
| superconduct $(10^2)$ | 1.012 (0.013) | 1.036 (0.014) | 1.015 (0.014) | 1.038 (0.013) |
| yprop_4_1 $(10^{-4})$ | 9.457 (0.530) | 9.481 (0.534) | 9.464 (0.532) | 9.474 (0.527) |
| abalone $(10^0)$ | 4.775 (0.066) | 4.788 (0.062) | 4.767 (0.068) | 4.759 (0.068) |
| **Regression MRR** | 0.72 | 0.43 | 0.65 | |
| **Classification (ROC AUC)** | | | | |
| credit | 0.857 (0.001) | 0.827 (0.001) | 0.857 (0.001) | 0.857 (0.001) |
| electricity | **0.950** (0.001) | 0.918 (0.001) | 0.948 (0.001) | 0.960 (0.001) |
| covertype | **0.933** (0.003) | 0.931 (0.003) | 0.932 (0.003) | 0.931 (0.003) |
| pol | 0.999 (0.000[†]) | 0.999 (0.000[†]) | 0.999 (0.000[†]) | 0.999 (0.000[†]) |
| house_16H | 0.951 (0.001) | 0.947 (0.001) | 0.951 (0.001) | 0.950 (0.001) |
| MagicTelescope | 0.931 (0.001) | 0.931 (0.001) | 0.931 (0.001) | 0.930 (0.001) |
| bank-marketing | 0.886 (0.001) | 0.886 (0.001) | 0.886 (0.001) | 0.886 (0.001) |
| MiniBooNE | 0.983 (0.000[†]) | 0.967 (0.000[†]) | 0.983 (0.000[†]) | 0.982 (0.000[†]) |
| eye_movements | 0.705 (0.003) | 0.709 (0.003) | 0.709 (0.003) | 0.718 (0.003) |
| Diabetes130US | 0.647 (0.001) | 0.647 (0.001) | 0.647 (0.001) | 0.647 (0.001) |
| jannis | 0.868 (0.001) | 0.867 (0.001) | 0.868 (0.001) | 0.867 (0.001) |
| default-of-credit-card-clients | 0.781 (0.002) | 0.779 (0.002) | 0.781 (0.002) | 0.780 (0.002) |
| Bioresponse | 0.861 (0.002) | 0.859 (0.002) | 0.860 (0.002) | 0.858 (0.002) |
| california | 0.967 (0.000[†]) | 0.962 (0.001) | 0.967 (0.000[†]) | 0.966 (0.000[†]) |
| heloc | 0.798 (0.002) | 0.798 (0.002) | 0.798 (0.002) | 0.798 (0.002) |
| **Classification MRR** | 0.68 | 0.41 | 0.70 | |

Table 7: Summary of quantile, uniform, and $k$-means binning on 33 real-world regression and classification datasets. Standard errors included in parentheses. Standard errors that round to 0.000 are marked [†] and are all $< 0.0005$.

## H.2 63-bin results

Table 8: Summary of quantile, uniform, and $k$-means binning on 18 real-world regression datasets, with bin budget $B = 63$. Standard errors included in parentheses.

| Dataset Name | Quantile | Uniform | $k$-means | Exhaustive |
|---|---|---|---|---|
| **Regression (MSE)** | | | | |
| cpu_act $(10^0)$ | 5.579 (0.203) | 5.416 (0.145) | **5.140** (0.126) | 5.092 (0.122) |
| pol $(10^1)$ | 3.483 (0.087) | 3.556 (0.087) | 3.421 (0.084) | 3.347 (0.088) |
| elevators $(10^{-6})$ | 4.936 (0.060) | 4.927 (0.061) | **4.859** (0.062) | 4.881 (0.068) |
| wine_quality $(10^{-1})$ | 4.142 (0.054) | 4.168 (0.049) | 4.114 (0.051) | 4.117 (0.053) |
| Ailerons $(10^{-8})$ | 2.529 (0.021) | 2.525 (0.021) | 2.510 (0.019) | 2.519 (0.021) |
| houses $(10^{-2})$ | 5.451 (0.042) | 6.030 (0.046) | **5.409** (0.046) | 5.292 (0.040) |
| house_16H $(10^{-1})$ | 3.419 (0.135) | 3.881 (0.130) | 3.398 (0.126) | 3.262 (0.132) |
| diamonds $(10^{-2})$ | **5.501** (0.021) | 5.810 (0.019) | 5.526 (0.021) | 5.459 (0.022) |
| Brazilian_houses $(10^{-3})$ | 6.507 (1.208) | 34.465 (0.850) | **2.222** (0.640) | 2.156 (0.576) |
| Bike_Sharing_Demand $(10^3)$ | 9.694 (0.046) | 9.693 (0.046) | 9.675 (0.044) | 9.694 (0.042) |
| nyc-taxi-green-dec-2016 $(10^{-1})$ | 1.767 (0.011) | 2.268 (0.007) | 1.763 (0.013) | 1.320 (0.016) |
| house_sales $(10^{-2})$ | 3.195 (0.024) | 3.322 (0.023) | 3.189 (0.022) | 3.206 (0.020) |
| sulfur $(10^{-4})$ | 5.089 (0.379) | 4.627 (0.389) | 4.589 (0.349) | 4.779 (0.383) |
| medical_charges $(10^{-3})$ | 7.365 (0.032) | 13.980 (0.126) | **6.737** (0.037) | 6.584 (0.035) |
| MiamiHousing2016 $(10^{-2})$ | 2.267 (0.031) | 2.312 (0.033) | 2.289 (0.029) | 2.309 (0.031) |
| superconduct $(10^1)$ | **9.931** (0.131) | 10.825 (0.131) | 10.193 (0.139) | 10.383 (0.133) |
| yprop_4_1 $(10^{-4})$ | 9.425 (0.535) | 9.432 (0.538) | 9.435 (0.531) | 9.474 (0.527) |
| abalone $(10^0)$ | 4.775 (0.067) | 4.837 (0.066) | 4.790 (0.065) | 4.759 (0.068) |
| **Regression MRR** | 0.59 | 0.39 | 0.85 | |

## H.3 XGB results

Table 9: Summary of binning methods $(B = 255)$ on three real-world representative regression datasets using XGBoost. Standard errors included in parentheses.

| Dataset Name | Quantile | Uniform | $k$-means | Exhaustive |
|---|---|---|---|---|
| **Regression (MSE)** | | | | |
| cpu_act $(10^0)$ | 5.047 (0.145) | 5.204 (0.166) | 5.454 (0.193) | 5.631 (0.229) |
| Brazilian_houses $(10^{-3})$ | 4.555 (0.913) | 19.303 (0.445) | **1.655** (0.407) | 1.639 (0.393) |
| superconduct $(10^2)$ | 1.003 (0.022) | 1.038 (0.022) | 1.014 (0.021) | 1.032 (0.021) |

