# OpenReview forum: "A Case for Library-Level $k$-Means Binning in Histogram Gradient-Boosted Trees"
_TMLR — Accepted by TMLR_

### Review · Reviewer_JvQy · 2025-08-08

**Summary Of Contributions:**

*Core Contributions*

1. Challenges GBDT Binning Heuristics: Identifies that quantile binning—the decades-old default in major libraries (XGBoost, LightGBM, CatBoost)—overlooks critical split points in skewed distributions.

2. Proposes K-Means Binning: Replaces quantile binning with k-means discretization (initialized by quantiles), proving it:
    Maximizes worst-case explained variance for L-Lipschitz functions (Lemma 1).
    Recovers label-relevant boundaries quantile binning misses.

3. Empirical Validation:

3.1 Real-world: On 18 regression datasets, k-means never underperforms quantile binning (5% significance) and achieves 55–66% MSE reductions on skewed data.

3.2 Synthetic: Controls for skew/modality/bin budget, showing >50–90% MSE gains when outliers exist and bins are scarce (e.g., GPU-friendly 16–64 bins).

3.3 Classification: Statistically tied with quantile binning (15 datasets).

3.4 Practical Implementation: Advocates for a bin_method=k-means flag in GBDT libraries—negligible overhead (3.5s for 10M rows), cacheable, and especially valuable for regression/GPU regimes.


*Key Strengths*

1. Theoretical Rigor: Lemma 1 elegantly links k-means to explained variance maximization. Explains why k-means wins: Isolates label-relevant extremes while quantile oversmooths tails.
2. Experimental Thoroughness:
33 real datasets + controlled synthetics covering skew, modality, bin budgets.
GPU-relevant tests (B= 16–64 bins) showing up to 90% MSE gains.
Validates results across libraries (scikit-learn, XGBoost).
3. Practical Impact:
Positions k-means as a "safe default": No losses, large upside.
Minimal overhead: One-time cost (≈ 3.5s for 10M rows) with cacheability.
4. Insightful Analysis:
Explains performance gap: Regression MSE is unbounded → outliers dominate errors; classification losses are bounded.
Synthetic suite isolates causality (e.g., outliers + low bins → 90% gains).

 *Key Weaknesses*

1. Limited Real-World Regression Wins: Only 3/18 datasets show significant k-means improvements. While "no losses" is strong, the upside is concentrated (Brazilian Houses dominates).
2. Unexplained Quantile Wins in Low-Bin Regime: Quantile binning outperformed k-means on 2 datasets at B= 63 bins. No hypothesis given (e.g., symmetric data?).

**Additional Comments:**

No

**Audience:**

Yes

**Audience Explanation:**

Key Audience Segments That Would Care:

1. GBDT Library Developers (e.g., XGBoost/LightGBM/CatBoost teams):
The proposal for a bin_method=k-means flag is a low-effort, high-impact change (negligible overhead, no losses, large regression upside).
GPU-optimization focus (16–64 bins) directly addresses hardware constraints in modern ML.
2. Practitioners Working on Skewed or High-Stakes Regression:
Finance, healthcare, or scientific domains often deal with heavy-tailed data (e.g., rare extreme events). The 55–90% MSE reductions on such data are compelling.
The "safe default" argument reduces adoption friction: no tuning needed, no downside.
3. Theoreticians Studying Efficient ML:
The explained-variance maximization proof (Lemma 1) offers a principled justification for k-means binning’s efficacy.
Synthetic experiments provide a controlled framework for studying binning’s impact on model performance.
4. Researchers in Tabular Data Learning:
Exposes a long-overlooked assumption in GBDTs: quantile binning’s hegemony.
Demonstrates that simple algorithmic swaps can rival complex innovations (e.g., neural nets for tabular data).

**Claims And Evidence:**

Yes

**Claims Explanation:**

Based on a comprehensive review of the theoretical proof, methodology, results, and synthetic diagnostics, the submission's core claims are strongly supported by accurate, convincing, and clear evidence. Here's a breakdown:


1. Claim: K-means binning outperforms quantile binning in regression, especially on skewed data/low-bin budgets.
Evidence:
55–66% MSE reduction on Brazilian Houses (highly skewed real-world data).
>50–90% MSE gains in synthetic tests with outliers + coarse bins (Figs. 1a, 2b).
No significant losses on 18 regression datasets (BH-corrected p>0.05).
8–66% gains under GPU-relevant bin budgets (B= 63).
Strength: Rigorous real-world benchmarks + controlled synthetics isolate causality.

2. Claim: K-means is a "safe default" (no downside, large upside).
Evidence:
0 significant losses vs. quantile across 33 datasets.
2–43% gains even in "favorable" settings (pure Gaussians, ample bins).
Negligible overhead (3.5s for 10M rows).
Strength: Statistical neutrality proven via 20 splits + BH correction.

3. Claim: K-means maximizes worst-case explained variance for L-Lipschitz functions.
Evidence:
Lemma 1 (paired-difference identity) rigorously proves the theoretical foundation.
Synthetic tests confirm k-means isolates label-relevant extremes (e.g., 20σ outliers).
Strength: Theory aligns with empirical gains on skewed data.

4. Claim: Classification performance is neutral.
Evidence:
MRR tie (0.70 vs. 0.68) on 15 classification tasks.
Max 0.2 pp AUC gap (statistically significant but practically negligible).
Strength: Bounded-loss explanation (log-loss ≤ ln2) justifies neutrality.

**Requested Changes:**

Strengthen the work:

1. Explain Quantile's Low-Bin Wins

Issue: Quantile significantly outperformed k-means on Superconduct (+ 2.5%) and Diamonds (+ 0.5%) at B= 63 bins (Sec 4.1.2).

Action: Add 1–2 sentences hypothesizing why.

2. Clarify MRR Calculation

Issue: Ambiguity in whether MRR is macro-averaged (per-dataset) or weighted by dataset size.

3. Justify Synthetic "100% Outliers" Label

Issue: Misleading term for exponentially distributed data (outliers imply rarity).

4. Address GPU Motivation Gap

Issue: Claims GPU relevance but lacks citation for quantile binning as a bottleneck.

Action: Cite some work for support.

---

> ### Author Response · Authors · 2025-08-29
> **Response to Reviewer JvQy**
>
> We greatly appreciate the thoughtful and specific feedback. Your comments helped us tighten several explanations. We now (i) explain why quantile can sometimes win in the low-bin regime, (ii) clarify exactly how MRR is computed, (iii) rename the confusing "100\% outliers" case to "all exponential," and (iv) briefly justify the GPU-relevant choice B=63. Details follow point-by-point below.
>
> > Explain Quantile's Low-Bin Wins
>
> We added a short hypothesis in the main text (where we report the B=63 results) and echoed it in the appendix. In particular, we add the following text:
>
> **Text Added (results section):**
> In Appendix E, we hypothesize two mechanisms for quantile's small wins in the low-bin regime: (1) k-means may spend a larger _proportion_ of a small bin budget isolating extremes that may not be label-relevant, and (2) coarser histograms also introduce greater variance, so edge-label alignment can occur by chance.
>
> **Text Added (appendix):**
> We hypothesize two mechanisms for quantile's two small wins in the low-bin regime. First, k-means can allocate the same *number* of bins to extreme values as at higher B, but when B is small those bins consume a much larger *proportion* of the total budget -- which can sometimes be counterproductive if the tails are not label-relevant. Second, quantile binning may simply win by chance -- in low-bin regimes, model performance becomes sensitive to dataset quirks, so accidental alignments between bin edges and label structure can drive small empirical wins.
>
> > 2: Clarify MRR Calculation
>
> We now specify that MRR is macro-averaged across datasets, with equal weight per dataset ("each dataset contributes one vote"). This change can be found in section 4.1.1.
>
> > 3: Justify Synthetic "100\% Outliers" Label:
>
> We totally see how that label is confusing. We've renamed the case to “all exponential” throughout the figures and text and clarified that it serves as a control drawn entirely from an exponential distribution. Despite its skew, it behaves much like the pure-Gaussian baseline, implying that $k$-means' advantage stems from a sparse tail + dense core together, not heavy-tailedness alone.
>
> > 4: Address GPU Motivation Gap:
>
> We added a brief explanation in Appendix E grounded in the LightGBM GPU paper of Zhang et al. (2017).
>
> **Text Added (Appendix E lead-in):**
>
> In LightGBM’s GPU implementation, Zhang et al. (2017, p.7, p.13)
> note that using fewer bins (6-bit packing with one reserved value, i.e., 63 usable bins) can increase available per-workgroup workload and reduce local memory requirements, both of which speed up training. They report significant speedups at this setting, so we rerun our real-world regression suite with B=63 to reflect the GPU-relevant regime and report those results below.

---

### Review · Reviewer_M918 · 2025-08-18

**Summary Of Contributions:**

This paper proposes a new histogram binning scheme for gradient boosted trees. Histogram binning of continuous features is used to reduce to complexity of learning splits on continuous features, Conventionally the bins used for this are uniform or quantile-based, with the latter often being the default. The authors question this choice and seek to find a more utile histogram binning scheme. To that end, they propose the use of 1-D K-means binning to bin continuous features. Specifically, they opt for k-means binning learned via Lloyd’s algorithm, initialized with quantiles instead of the optimal dynamic programming algorithm for performing 1-D K-means.

The utility of the proposed scheme is demonstrated on a variety of real regression and classification datasets. Additional studies on synthetic datasets are used to determine the settings wherein K-means binning is more performant.

**Strengths:**

The paper proposes a simple, but novel idea, to improve the performance of Gradient Boosted Decision Trees (GDBTs), a widely used machine learning technique. The performance of K-means histogram binning was benchmarked against quantile and uniform binning across a wide set of regression and classification datasets. Each of the methods was accompanied by extensive hyperparameter tuning and cross-validation. This lends credence to the presented findings, especially given the use of statistical tests to ascertain if the presented findings are statistically significant.
The study on synthetic data is also very welcome, as the authors have sought to demonstrate why K-means binning is so much more performant on some real-world datasets. Moreover, the choice of model for synthetic data is not overly simplistic, which lends further credence to their findings.

**Weaknesses:**

I do not believe there are any glaring weaknesses in this work but there are several questions that should be answered in a subsequent revision.

1.	The added cost of K-means binning lies in the pre-training stage where continuous features are binned into histograms. This unsurprisingly is more expensive than quantile or uniform binning – does this cost become prohibitive for datasets with a lot of features? If yes, what is the upper bound on the feature space for K-means binning to be a viable alternative to quantile binning?
2.	What is the motivation for using mean reciprocal rank (MRR) as a metric? It is also not clear how this is computed for each of the presented sets of results.
3.	What measure of skew did you use in Appendix B?

**Audience:**

Yes

**Audience Explanation:**

The proposed K-means binning histogram binning scheme is a simple modification to GDBTs, which are a commonly used machine learning technique. In the case of improved performance, this could be easily adopted by machine learning practitioners.

**Broader Impact Concerns:**

I do not have any broader impact concerns with this work.

**Claims And Evidence:**

Yes

**Claims Explanation:**

K-means histogram binning was benchmarked against quantile and uniform binning across a wide set of regression and classification datasets. The results were further evaluated using statistical significance tests to ascertain which of the data points are meaningful performance improvements. Studies on synthetic data are also used to demonstrate the data regime in which the proposed histogram binning scheme is more performant than uniform and quantile histogram binning.

**Requested Changes:**

**Critical changes:**

1.	It is not clear what “library-level” on Page 3 means.
2.	Given the extensive hyperparameter tuning conducted, the reported results should have accompanying error bounds (or some measure of variability). This will also solidify the conclusions made about the statistical significance of the difference in performance across binning schemes.
3.	Specify the measure of skew that you used in Section B (Tables 2a and 2b).
4.	Motivating the design choices and parameter specification in Algorithm 1 is important, as the conclusions made from the study on synthetic data are predicated on this data being somewhat realistic.

**Suggested changes:**

1.	Clarifying what “pp” means on Page 2 to improve clarity.
2.	Results with the exhaustive split search are reported in Tables 1, 2, 5 and 6, presumably as a baseline. However, those results are not discussed anywhere in the text.
3.	What is the intuition behind K-means binning with k-means++ initialization performing worse than quantile-based initialization?

---

> ### Author Response · Authors · 2025-08-29
> **Response to Reviewer M918**
>
> We truly appreciate your thorough comments and suggestions -- they helped us clean up and clarify several sections. We respond to each point in order below, with corresponding updates to our paper noted. Due to the 5000 character limit, we have split our response into two parts.
>
> > The added cost of K-means binning lies in the pre-training stage where continuous features are binned into histograms. This unsurprisingly is more expensive than quantile or uniform binning – does this cost become prohibitive for datasets with a lot of features? If yes, what is the upper bound on the feature space for K-means binning to be a viable alternative to quantile binning?
>
> Since all three binning methods are applied to each feature individually, they scale linearly with the number of features. Importantly, GBDTs **also** scale linearly with respect to the number of features -- and since training GBDTs themselves is slower at every row count tested, it will remain slower as the number of features grows.
>
> When memory -- not compute -- becomes the bottleneck, packages can turn to streaming 1-D k-means algorithms that require only O(B) memory, such as the one described in Sculley, 2010.
>
> We've updated Section 5 to note these facts.
>
> > What is the motivation for using mean reciprocal rank (MRR) as a metric? It is also not clear how this is computed for each of the presented sets of results.
>
> We include mean reciprocal rank (MRR) as a scale-free summary over the very heterogeneous set of datasets we benchmark. Absolute test errors span more than ten orders of magnitude (e.g., MSE $\approx 10^3$ on Bike_Sharing_Demand vs $10^{-8}$ on Ailerons), so averaging raw scores would be dominated by the largest-scale problems. Rank-based metrics avoid that pitfall.
>
> As for how we calculate it, we now specify that MRR is macro-averaged across datasets, with equal weight per dataset ("each dataset contributes one vote"). Both changes can now be found in section 4.1.1.
>
> > What measure of skew did you use in Appendix B?
>
> We follow SciPy's default Fisher-Pearson sample skewness (stats.skew) with default values (bias = True). For a robustness check we recalculated with bias = False (unbiased version) and found differences at the $10^{-3}$ level; all values printed in Table 2 remain the same due to rounding. Because the correction has no practical effect, we retain the widely used default and now state this choice explicitly in the manuscript.
>
> > It is not clear what “library-level” on Page 3 means.
>
> We used "library-level" to mean "integrated directly into the package itself"; i.e. users only need to flip bin_method="k-means" just as they choose objective="reg:squarederror". We have noted this via a footnote on page 2, where library-level is defined.
>
> > Given the extensive hyperparameter tuning conducted, the reported results should have accompanying error bounds (or some measure of variability). This will also solidify the conclusions made about the statistical significance of the difference in performance across binning schemes.
>
> Great point. We've now added an Appendix H, which adds SE to all experimental tables. In Section 4.1.1, we've also added the following text to point readers towards that appendix:
>
> **Text added (4.1.1):**
>
> In Appendix H we mirror all experimental tables -- including those in Appendices E and F -- and add the standard error across random seeds.
>
> > Motivating the design choices and parameter specification in Algorithm 1 is important, as the conclusions made from the study on synthetic data are predicated on this data being somewhat realistic.
>
> Agreed. We deliberately kept the synthetic generator as *simple* and *transparent* as possible while still reproducing the two properties we wanted to probe: *multi-modality* and *heavy, label-relevant tails*. We now clarify this within the paper via a short rationale paragraph in Appendix G, which contains the algorithm for synthetic data generation.
>
> **Text added (Appendix G)**
>
> We motivated this algorithm as a combination of the simplest building blocks that yield the distributional features we wanted to probe (multi-modality and label-relevant outliers).
>
> First, a Gaussian mixture
> with n_modes components creates a controllable number of modes. We fix dist=4 so that adjacent
> modes share $\approx 2.5\%$ of their mass -- enough to avoid empty bins
> while keeping the peaks distinct.
>
> Second, we standardize each feature to equalize scale before any outliers
> are introduced.
>
> Third, an exponential distribution of size $\beta$ applied to a p_out fraction of samples generates controllable outlier magnitude and frequency.
>
> Finally, a simple linear target ($y_i = \sum_{j} X_{ij} + \epsilon_i$) ensures the exponential tails are label-relevant (i.e. extreme $X$ maps to extreme $Y$).

---

> > ### Author Response · Authors · 2025-08-29
> > **Response to Reviewer M918 (Part 2)**
> >
> > > Clarifying what “pp” means on Page 2 to improve clarity.
> >
> > Totally fair. We now define “pp” at first use -- writing “percentage-points (pp)” -- and use the abbreviation afterwards for brevity.
> >
> > > Results with the exhaustive split search are reported in Tables 1, 2, 5 and 6, presumably as a baseline. However, those results are not discussed anywhere in the text.
> >
> > We've added two paragraphs -- one in 4.1.1, and one in 4.1.2, that describe and analyze the exhaustive search column. Since our study focuses on practical binning methods, we do not include exhaustive search in the per-dataset significance tests or in the MRR summaries. Instead, it serves as an un-binned reference that lets readers gauge the accuracy degradation attributable to binning.
> >
> > **Text added (4.1.1):**
> >
> > In all real-world benchmark tables, we also list an "exhaustive search" column, which checks all possible split thresholds without using binning. Because that variant enumerates all possible split thresholds (O(N) per node), it is impractically slow on large datasets but serves as a practical upper-bound reference to quantify the accuracy degradation attributable to binning itself.
> >
> > **Text added (4.1.2):**
> >
> > Across the full regression suite, $k$-means performs $\geq 0.3\%$ worse than the exhaustive split method on just three datasets; of those, quantile performs statistically significantly worse than $k$-means on two. This suggests that $k$-means retains almost all of the predictive value of exhaustive splitting when extra cuts add little value, yet still recovers key tail-driven split-points that quantile overlooks.
> >
> > > What is the intuition behind K-means binning with k-means++ initialization performing worse than quantile-based initialization?
> >
> > We realized the wording in the original draft was ambiguous in two ways and appreciate you flagging it:
> >
> > 1. “Quantile-based initialization” referred to k-means whose centroids are initialized at empirical quantiles, not to the quantile-binning baseline.
> > 2. Our pilot runs showed no accuracy drop for k-means++; the only difference was runtime. k-means++ incurs an extra O(nk) initialization pass (Arthur & Vassilvitskii 2007), which increased preprocessing time without yielding a measurable performance gain.
> >
> > Because the added cost provided no benefit, we chose the lighter quantile seeding. We have rephrased Section 5 to make this trade-off explicit.
> >
> > **Text changed (section 5):**
> >
> > We also experimented with $k$-means++ seeding (Arthur
> > & Vassilvitskii, 2007), but saw no meaningful accuracy improvement over $k$-means with quantile seeding. Since the extra $O(nk)$ initialization pass in $k$-means++ made preprocessing substantially slower, we decided to use quantile seeding throughout.

---

> > > ### Comment · Reviewer_M918 · 2025-09-03
> > > **Minor comments**
> > >
> > > I would like tot hank the authors for taking on board the suggestions from all the reviewers. I believe that the edits have significantly strengthened the manuscript.
> > >
> > > I have a minor comment regarding the text added to Section 4.1.2: what is the unit used to when saying that "$k$-means performs $\geq 0.3%$ worse than the exhaustive split method on just three datasets". If it refers to MSE, then this is somewhat confusing because the scales of MSE varies widely across the various datasets.

---

> > > > ### Author Response · Authors · 2025-09-03
> > > > **Response to Minor comments by Reviewer M918**
> > > >
> > > > We are very happy to hear that our revisions alleviated your concerns.
> > > >
> > > > As for your additional minor comment, we apologize -- that was a typesetting typo in our OpenReview response (but not in the manuscript). In particular, we write in Section 4.1.2 that **"Across the full regression suite, $k$-means performs $\mathbf{\geq 0.3}$% worse than the exhaustive split method on just three datasets"**.
> > > >
> > > > As with the rest of the paper, we compare $k$-means with exhaustive split in a unitless manner (percentage relative MSE), such that $\frac{(MSE_{k-\text{means}} - MSE_{\text{exhaustive}})}{MSE_{\text{exhaustive}}} \geq 0.003$ on just three datasets.

---

### Review · Reviewer_DMQN · 2025-08-19

**Summary Of Contributions:**

The authors show that using K-means to select bins in histogram gradient-boosted trees leads to significantly improved accuracy for regression on datasets with a large (but not too large) set of outliers, or when the number of bins is quite limited.  They also show that performance is very similar to the state-of-the-art binning technique (quantile binning) when there are very few outliers or when the number of bins is less constrained.  Furthermore, it is shown that on classification tasks, K-means binning also performs very similarly to quantile binning.  Finally the authors' approach is efficient in terms of computing time.  Extensive experiments on OpenML datasets and carefully constructed synthetic datasets are used to justify these claims.  The authors also show that using K-means maximizes a tight lower bound on the explained variance of the target variable assuming a noisy L-Lipschitz relationship between features and target.

In summary, the authors have proposed a practical method for improving histogram gradient-boosted trees.  The evidence is clear and convincing, and the paper is well written.  I am convinced that K-means should be included as a binning technique for Gradient-Boosted Decision Trees in widely-used machine learning libraries.

**Audience:**

Yes

**Audience Explanation:**

The Gradient-Boosted Decision Trees approach is successful and widely used.  Binning the feature values is an excellent way to speed up the algorithm for large datasets.  The authors show their approach achieves significant improvements over the current state-of-the-art in relevant settings, without performance loss in other settings.  Furthermore, the authors' approach is efficient in terms of computing time, and does not create significant overhead compared to existing binning approaches.  I am convinced that K-means should be included as a binning technique for Gradient-Boosted Decision Trees in widely-used machine learning libraries.

**Claims And Evidence:**

Yes

**Claims Explanation:**

Table 1 provides a comparison of the different binning techniques for a wide range of OpenML datasets.  Figures 1 and 2 show the performance gain of using K-means over quantile binning for carefully constructed datasets where the outlier properties, and the number of bins are controlled.  These experiments show precisely when K-means leads to huge improvements over quantile binning and clear, intuitive explanations for why this is the case accompany the empirical results.

The theoretical motivation section provides a clear proof for the fact that K-means maximizes a tight lower bound on the explained variance of the target variable.  While an analysis of how this affects the performance of gradient-boosted trees is not included, I view this result on explained variance as a good motivation for the work.

**Requested Changes:**

The following changes are recommended, and none of them (other than correcting critical typos) are essential in my opinion.

1. It would be nice to include an explanation and example of the gradient-boosted decision tree algorithm.  It is more reader-friendly when a paper is self-contained.

2. Explain what atomic unit means.

3. Explain the notation $\text{E}_j[]$ and $\text{Var}_j[]$.

4. Explain the meaning of K-means on a probability distribution in the theoretical motivation.  Many readers are likely to only have seen K-means work on a set of data points.

5. Explain why K-means minimizes the expected variance of $X$ or give a reference.

6. Indicate what version of ROC-AUC you are using when there are multiple classes.

7. The regression MRR's in Table 1 do not match the MRR's reported in Discussion 4.1.2.  I believe this is a simple typo.

---

> ### Author Response · Authors · 2025-08-29
> **Response to Reviewer DMQN**
>
> We very much appreciate the detailed feedback. Your feedback helped us clarify notation and provide important background information to readers. We’ve revised the manuscript accordingly and respond point-by-point below.
>
> > It would be nice to include an explanation and example of the gradient-boosted decision tree algorithm. It is more reader-friendly when a paper is self-contained.
>
> Agreed. We've added a short description of GBDTs in section 1.
>
> **Text added:**
>
> In *formal terms*, GBDT algorithms wish to minimize an arbitrary twice-differentiable loss function by maintaining an ensemble-based predictor $F_t$ comprised of a series of CART trees. At each round t it calculates first and second-order statistics
>
> $$
> \begin{equation*}
>     g_i = \frac{\partial L(y_i, F_{t-1}(x_i))}{\partial F_{t-1}(x_i)}
>     \qquad
>     h_i = \frac{\partial^2 L(y_i, F_{t-1}(x_i))}{\partial F_{t-1}(x_i)^2}
> \end{equation*}
> $$
>
> and then fits a simple CART tree to maximize a split-gain criterion. The optimal weight for leaf $l$ is calculated as
> $$
> w_l = -\frac{\sum_{i \in l} g_i}{\lambda + \sum_{i \in l} h_i}
> $$
>
> and the model is updated as $F_t(x) = F_{t-1}(x) + \eta w_{leaf_t(x)}$. Parameters $\eta$ and $\lambda$ correspond to the learning rate and L2 regularization respectively, and are chosen by the user prior to training.
>
> > Explain what atomic unit means.
>
> We intended to use "atomic unit" to mean that "all values in a given bin are treated as equivalent". We decided to simply replace all uses of the first with the second to reduce confusion.
>
> > Explain the notation $\mathbb{E}_j[]$ and $\operatorname{Var}_j[]$.
>
> We have added a short explanation of the notation prior to using it. In particular, we write $E_j[g_j] = \sum_{j=1}^K \pi_j g_j$ and $Var_j(g_j) = \sum_{j=1}^K \pi_j \bigl(g_j - E_j[g_j]\bigr)^2$ as the expectation/variance of some function of g across bins under $\{\pi_j\}$.
>
> > Explain the meaning of K-means on a probability distribution in the theoretical motivation. Many readers are likely to only have seen K-means work on a set of data points.
>
> We provide a short explanation of distributional $k$-means and justification for why it minimizes the expected within-bin variance of X. It can be found below or at the beginning of section 3.
>
> **Text added:**
>
> Throughout this section we work in the *population* (distributional) setting: a $K$-binning is a measurable partition $B=\{B_1, B_2, ... B_K\}$ of the feature domain, and we write $\pi_j = \mathbb{P}(X \in B_j)$. Given centers $\{c_1, ..., c_K\}$ the $k$-means quantization (binning) objective is to minimize
> $$
> \sum_{j=1}^K \pi_j \mathbb{E}[(X - c_j)^2 | X \in B_j]
> $$
>
> For any fixed partition $B$, the objective is minimized with respect to the centers when $c_j = \mathbb{E}[X | X \in B_j]$; substituting yields
> $$
> \sum_{j=1}^K \pi_j \operatorname{Var}(X | X \in B_j)
> $$
>
> Because the optimization is taken jointly over all measurable $B$ *and* the centers $c_1, ..., c_K$, population-level $k$-means is therefore *exactly* the problem of minimizing the expected within-bin variance across all $K$ partitions. This mirrors the familiar sample version, where $k$-means minimizes the within-cluster variance of the observed data points (LLoyd, 1982).
>
>
> > Explain why K-means minimizes the expected variance of $X$ or give a reference.
>
> The paragraph above discusses this as well.
>
> > Indicate what version of ROC-AUC you are using when there are multiple classes.
>
> All classification tasks are binary, and as such we report the classic binary ROC-AUC metric. We have now noted this throughout the paper, including in the results section.
>
> > The regression MRR's in Table 1 do not match the MRR's reported in Discussion 4.1.2. I believe this is a simple typo.
>
> Yep, you're absolutely right -- we've fixed the typo. We appreciate your diligence in finding and pointing that out to us. The updated text can be found below, or in section 4.1.2.
>
> **Text changed**:
>
> Although $k$-means attains a slightly lower MRR compared to quantile binning (0.65 vs. 0.72), it is never statistically outperformed by either quantile or uniform binning at the 5% significance level across all evaluated datasets.

---

### Author Response · Authors · 2025-08-29
**Global Revision Notes**

We would like to thank all three reviewers and the Action Editor for their time and care invested in evaluating our work. We have revised the paper accordingly and responded point-by-point to every comment. In addition, we fixed two minor, unsolicited issues that do not affect any results:

(i) a small algebraic typo in the linear-model proof (Appendix A), and

(ii) a misstatement in the abstract (“four” $\rightarrow$ “three” wins) that now matches Table 1 and Section 4.1.2. No figures, numbers, or conclusions changed -- only the wording was corrected for consistency.

We believe these changes fully resolve the reviewers’ concerns and improve the clarity of the paper.

---

### Decision · Action_Editor_M9Zx · 2025-09-22

**Recommendation:** Accept as is

**Audience:**

Yes

**Audience Explanation:**

Gradient-boosted trees are widely used and this technique should be of interest to several members of the TMLR audience, both from an algorithmic/theoretical standpoint and from a practical adoption standpoint.

**Claims And Evidence:**

Yes

**Claims Explanation:**

Histogram-based binning is commonly used to speed up split finding in gradient-boosted trees. The authors propose a new histogram binning scheme based on 1-D K-means via Lloyd's algorithm, initialized with quantiles. Theoretically, the authors show that using K-means maximizes a tight lower bound on the explained variance of the target variable. Empirically, the performance of the proposed approach was benchmarked against quantile and uniform binning across a wide set of regression and classification datasets. The methods was evaluated based on careful hyperparameter tuning and cross-validation, lending credence to the findings. Additional studies on synthetic datasets are used to explain why the proposed approach excels on some real-world datasets. There is consensus among all reviewers that the (short) theoretical analysis combined with an extensive empirical evaluation provide clear and convincing evidence of the paper's contributions and of the practical potential impact of the proposed technique.